# Health-related quality of life and estimation of the minimally important difference in the Functional Assessment of Cancer Therapy-Endocrine Symptom score in postmenopausal ER+/HER2- metastatic breast cancer with low sensitivity to endocrine therapy

**Yuichiro Kikawa** [1]*, **Yasuhiro Hagiwara**[2], **Tomomi Fujisawa**[3], **Kazuhiro Araki** [3,4], **Takayuki Iwamoto**[5], **Takafumi Sangai**[6], **Tadahiko Shien**[5], **Shintaro Takao**[7], **Reiki Nishimura**[8], **Masato Takahashi** [9], **Tatsuya Toyama**[10], **Tomohiko Aihara**[11], **Hirofumi Mukai**[12], **Naruto Taira**[5]

**1** Department of Breast Surgery, Kansai Medical University, Hirakata-city, Osaka, Japan, **2** Department of Biostatistics, The University of Tokyo, Tokyo, Japan, **3** Department of Breast Oncology, Gunma Prefectural Cancer Center, Ota-city, Gunma, Japan, **4** Department of Breast and Endocrine Surgery, Hyogo College of Medicine, Nishinomiya-city, Hyogo, Japan, **5** Department of Breast and Endocrine Surgery, Okayama University Hospital, Okayama-city, Okayama, Japan, **6** Department of Breast and Thyroid Surgery, Chiba University Hospital, Chiba-city, Chiba, Japan, **7** Department of Breast Surgery, Hyogo Cancer Center, Akashi City, Hyogo, Japan, **8** Department of Breast Oncology, Kumamoto Shinto General Hospital, Kumamoto-city, Kumamoto, Japan, **9** Department of Breast Surgery, NHO Hokkaido Cancer Center, Sapporo-city, Hokkaido, Japan, **10** Department of Breast Surgery, Nagoya City University Hospital, Nagoya-city, Aichi, Japan, **11** Breast Center, Aihara Hospital, Mino-city, Osaka, Japan, **12** Department of Breast and Medical Oncology, National Cancer Center Hospital East, Kashiwa-city, Chiba, Japan

* kikaway@hirakata.kmu.ac.jp

## Abstract

### Background

The HORSE-BC study previously demonstrated that second-line endocrine therapy (ET) for patients with acquired endocrine-resistant metastatic breast cancer (MBC) still provided a clinically meaningful benefit. Herein, we investigated the health-related quality of life (HRQOL) in the HORSE-BC study.

### Methods

Patients with acquired endocrine-resistant MBC who were scheduled for second-line ET were recruited. The HRQOL was assessed at baseline, and 1 and 3 months after second-line ET initiation. To investigate the minimally important difference (MID) in the Functional Assessment of Cancer Therapy-Endocrine Symptoms (FACT-ES), we evaluated the means and standard deviations for the distribution-based method, and differences in the change in HRQOL for the anchor-based method. We also investigated the association between FACT-ES total scores and clinical benefit.

**Data Availability Statement:** The data underlying the results presented in the study are available from Comprehensive Support Project for Oncological Research of Breast Cancer (CSPOR-BC). Please contact us through the web-site (https://cspor-bc.or.jp/) for the relevant data.

**Funding:** This study was funded by AstraZeneca. The funder had no role in study design, data collection and analysis, decision to publish, or preparation of the manuscript.

**Competing interests:** YK received honoraria from Eisai, Novartis, Pfizer, Lilly, Taiho, and Chugai, outside the submitted work. MT received grants from Kyowa Hakko Kirin and Taiho, personal fees from AstraZeneca, Eisai, Eli Lilly, and Pfizer, outside the submitted work. TT received grants and personal fees from Chugai, Eisai, Novartis, Takeda, Nippon Kayaku, Pfizer, Lilly, and Daiichi Sankyo, and personal fees from AstraZeneca, and grants from Taiho and Kyowa Kirin, outside the submitted work. HM received honoraria from Pfizer, Takeda, Daiichi Sankyo, and Taiho, and grants from the Japanese government, Daiichi Sankyo, and Pfizer, outside the submitted work. The remaining authors have no conflicts of interest to disclose. This does not alter our adherence to PLOS ONE policies on sharing data and materials.

**Abbreviations:** AI, aromatase inhibitor; BCS, breast cancer subscale; CBR, clinical benefit rate; CDK 4/6, cyclin-dependent kinase 4/6; CI, confidence interval; ESS, endocrine symptom subscale; ET, endocrine therapy; emotional well-being; FACT-ES, Functional Assessment of Cancer Therapy-Endocrine Symptoms; FWB, functional well-being; HER2, human epidermal growth factor receptor 2; HR, hormone receptor; HRQOL, health-related quality of life; MBC, metastatic breast cancer; MID, minimally important difference; PFS, progression-free survival; PRO, patient-reported outcomes; PWB, physical well-being; SEM, standard error of measurement; SFWB, social and family well-being; SSQ, subjective significant questionnaire; TOI, trial outcome index.

## Results

Overall, 56 patients were enrolled. Of these, 47 were analyzed. When defined as 1/3 standard deviation estimates based on the distribution method, the calculated MID was 5.9. The MIDs of the FACT-ES total scores based on the anchor method were 7.7 for decline and 4.1 for improvement. The MID decline proportions were 6.1% and 14.7% lower in patients who experienced clinical benefits than in those who did not at 1 and 3 months, respectively. The ratios of MID improvement in patients who experienced clinical benefits were 18.3% and 3.2% higher, respectively; the mean change in the FACT-ES total score from baseline improved in patients who experienced clinical benefits.

## Conclusions

Maintaining the HRQOL as determined by FACT-ES may be associated with clinical benefits in patients with acquired endocrine-resistant MBC treated with ET.

## Background

Endocrine therapy (ET) (plus targeted therapy) is recommended as an initial therapy for patients with hormone receptor (HR)-positive and human epidermal growth factor receptor 2 (HER2)-negative metastatic breast cancer (MBC) without visceral crisis according to the Hortobagyi algorithm and some guidelines [1–3]. Furthermore, subsequent ET should be continued even after patients develop resistance to primary ET because it will not only offer some disease control but will also cause fewer severe adverse events than with chemotherapy. Currently, the combination of ET plus cyclin-dependent kinase (CDK) 4/6 inhibitors is the standard second-line therapy for patients who have developed resistance to initial ET alone, based on the results of trials showing prolonged survival [4–6]. However, the sensitivity to subsequent ET (plus targeted therapy) varies among patients, and sometimes, choosing between sequential ET and chemotherapy is difficult, especially for patients assumed to have a low sensitivity to ET, such as those who relapsed during the first 2 years of adjuvant ET or had disease progression within 6 months of first-line ET as described in the ESO-ESMO Guidelines [2]. Therefore, we conducted a multicenter prospective observational cohort study (HORSE-BC study) including patients with HR-positive and HER2-negative MBC who were considered as having a low sensitivity to initial ET before CDK 4/6 inhibitors were available in Japan. Consequently, we concluded that second-line ET might be a viable treatment option for postmenopausal patients with MBC with low sensitivity to the initial ET [7].

In addition to better clinical response and delayed progression, maintaining health-related quality of life (HRQOL) is critical for patients with incurable MBC. HRQOL is defined as the quantitatively evaluated physical and mental health influenced by a disease or treatment [8, 9]. It is a multidimensional concept comprising fundamental domains, such as functional, physical, psychological, and social domains. To evaluate HRQOL, it is important to use patient-reported outcomes (PRO) as a subjective assessment.

There are studies on the HRQOL between treatments, which focused on the score differences and variations obtained by questionnaires. However, the methods of analysis and handling of missing data have not been standardized [10]. Further, whether statistically significant differences are clinically important may be unclear. Therefore, it is crucial to interpret the clinically meaningful differences, which are inherent in the score values. As an indicator for

interpreting these differences, the minimally important difference (MID) of the HRQOL score is an essential concept in cancer clinical research [11, 12]. Although several articles estimating the MIDs of HRQOL scales used in breast cancer have been published [13–15], there are no studies on the Functional Assessment of Cancer Therapy-Endocrine Symptoms (FACT-ES) [16] in ET for breast cancer. Furthermore, the effect of ethnicity, cultural background, and treatment interventions on MID values in the already estimated HRQOL score such as FACT-Breast (B) is unclear [17].

The main objectives of the analysis of the HRQOL in the HORSE-BC study were as follows: First, to clarify the clinically significant scores of the HRQOL as MIDs associated with ET for MBC. Second, to determine the effect of secondary ET on the HRQOL in postmenopausal patients with MBC that was less sensitive to primary ET, using the proportion of patients without a decline in the MID.

## Methods

### Study design and participants

The HORSE-BC study was a multicenter prospective observational study which aimed to evaluate the efficacy and safety of secondary ET among postmenopausal patients with ER-positive and HER2-negative MBC who had a low sensitivity to initial ET (the UMIN Clinical Trial Registry, UMIN000019556). Patients who received previous ET as continuous post-operative adjuvant therapy with recurrence within five years after ET initiation or patients who progressed within nine months after initial ET initiation for MBC were included in the study. The other main inclusion criteria were: An Eastern Cooperative Oncology Group performance status scores of 0–1 and either no previous chemotherapy for MBC or no pre- or post-operative chemotherapy within the past six months. We expected that secondary ET for MBC with a low sensitivity to ET will yield a clinical benefit rate (CBR) of at least 30% using newer endocrine agents; the expected CBR was 50%. Overall, 56 patients were enrolled between February 2016 and January 2017, of whom 49 were analyzed. The study results have been described previously [7].

### HRQOL assessment

HRQOL was assessed at baseline, 1 month, and 3 months after initiation of second-line ET using the Japanese version of the FACT-General (G), FACT-B, and FACT-ES. The FACT-G measures the general HRQOL associated with cancer using 27 items (7 for physical well-being [PWB], 7 for social and family well-being [SFWB], 6 for emotional well-being [EWB], and 7 for functional well-being [FWB]), and the FACT-B has 10 additional items on the breast cancer subscale (BCS) that are more specific to women with breast cancer [17, 18]. The FACT-ES was designed for use with the FACT-B. It is comprised of both 19 items on the endocrine symptom subscale (ESS) and 27 items for the FACT-G, with a possible maximum score of 184 [16]. Each item or question on the FACT-G, B, and ES has response choices ranging from 0 ("not at all") to 4 ("very much"), which can be converted to total scores and trial outcome index (TOI), and are often used as the main outcome. The FACT-G TOI is the sum of PWB and FWB scores, with a maximum score of 56. The FACT-B TOI is the sum of PWB, FWB, and BCS scores with a maximum score of 96. The FACT-ES TOI is the sum of PWB, FWB, and ES scores with a maximum score of 132.

### Subjective significant questionnaire

To identify MIDs, we provided six questions corresponding to PWB, SFWB, EWB, FWB, ES, and general health as the subjective significant questionnaire (SSQ) [13] at 1 month and 3

months after the start of second-line ET along with the other HRQOL questionnaires. The SSQs include Likert scales for seven questions written as follows: Since you decided to participate in this study, (A) your physical condition is. . ., (B) your social or relationship with your family people are. . ., (C) your degree of anxiety is . . ., (D) your social activities are. . ., (E) your menopausal symptoms are. . ., and (F) your general health status is. . . The choices after each question are as follows: (1) very much better, (2) moderately better, (3) a little better, (4) about the same, (5) a little worse, (6) moderately worse, and (7) very much worse.

## Data analysis

We analyzed data from patients with baseline HRQOL assessments. The completion rates for each HRQOL score at each time point were calculated as the number of patients who had each HRQOL score divided by the number of patients included in the HRQOL analysis. We summarized the obtained HRQOL scores at each time point with mean and standard deviation ($\sigma$). We did not impute missing HRQOL scores and analyzed only the available scores.

We estimated MIDs of the FACT-G total score and TOI, FACT-B total score and TOI, FACT-ES total score and TOI, and ESS. To estimate MIDs based on the distribution method [14], we calculated 1/2 $\sigma$, 1/3 $\sigma$, and the standard error of measurement (SEM) of each score. The SEM was defined as $\sigma (1 - \text{rel})^{1/2}$, where rel was the reliability of each scale. We used Cronbach's $\alpha$ as a reliability measure in this study. We summarized MID estimates at baseline, 1 month, and 3 months using a weighted mean with the number of scores at each time point as a weight.

Responses to the SSQ and the corresponding HRQOL score change values [13] were used to estimate MIDs based on the anchor method [14]. We used SSQ (E) for the ESS, whereas SSQ (F) was used for the remaining scores. We compared the mean change in the HRQOL scores of patients with declined (i.e., much worse, moderately worse, or a little worse) and improved (i.e., very much better, moderately better, or a little better) SSQ responses to those with stable (i.e., about the same) SSQ responses using the generalized estimating equation method to account for the correlation between HRQOL scores at 1 month and 3 months. We also fitted a dose response model assuming a linear spline in the association between the SSQ responses and changes in HRQOL scores using the generalized estimation equation method. Using the linear spline model with 1 knot at a stable response to the SSQ, we estimated a mean change in the HRQOL scores corresponding to a moderate and little change in SSQ.

We also investigated the association between the FACT-ES total score and clinical benefit (defined as complete response, partial response, or stable disease for 24 weeks; the primary endpoint in the HORSE-BC study). We described cumulative distribution functions at each time point according to the clinical benefit status and compared the proportions of patients who experienced changes in MIDs between the clinical benefit status using the Fisher's exact test. We also compared mean profiles of changes in the FACT-ES total scores according to the clinical benefit status using the generalized estimating equation method. All statistical analyses were performed using the SAS software (version 9.4; SAS Institute Inc., Cary, NC, USA). All p-values were two-sided. Differences were considered statistically significant at $p < 0.05$, without a multiplicity adjustment. We did not make small-sample corrections to the sandwich variance estimator in generalized estimating equation analyses because more than 40 patients were included and HRQOL scores from only two visits were analyzed.

## Ethics approval and consent to participate

All procedures were performed in accordance with the Helsinki declaration and the institutional review board at each study site approved the final protocol (The names of the ethics

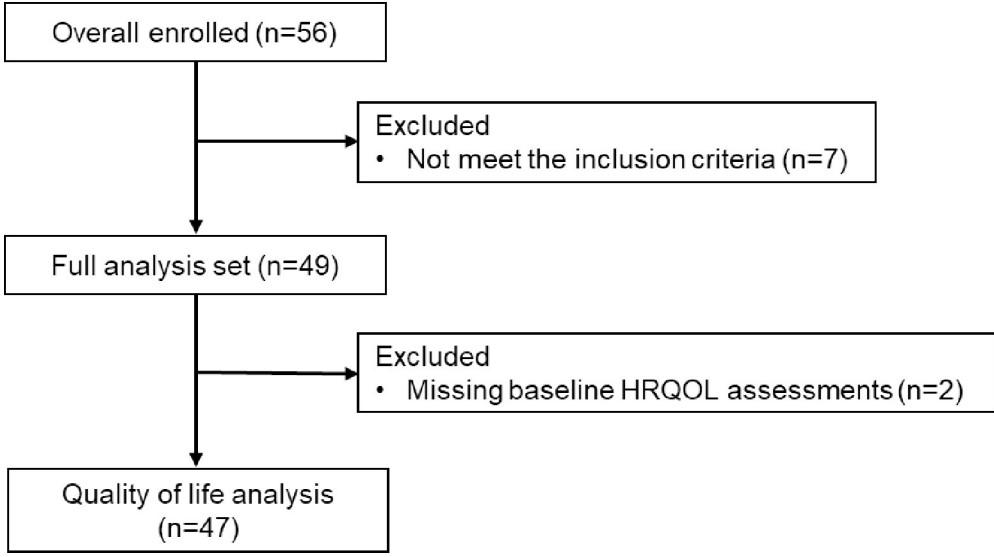

**Fig 1. Patients' flow chart.** HRQOL, health-related quality of life.

committees/institutional review boards were shown in the S1 File). Written informed consent was obtained from all study participants. The ethics committees that reviewed and approved this study are listed in the S1 File.

## Results

### Patients

Of the 56 patients enrolled in the HORSE-BC trial, 7 were excluded based on the inclusion criteria and 2 were excluded due to missing baseline HRQOL assessments. Finally, 47 patients were included in the HR-QOL analysis (**Fig 1**). The patient characteristics of the QOL/PRO analysis population are shown in **Table 1**.

### Distribution-based MID estimates

HRQOL scores and MID estimates based on the distribution-based methods are shown in **Table 2**. The completion rates were higher than 90% in all scores at all time points. For all scores, the smallest distribution-based MID estimates summarized by the weighted mean were 1/3 $\sigma$, followed in order by SEM and 1/2 $\sigma$. The distribution-based MID estimates for FACT-B total scores were 8.8 by 1/2 $\sigma$, 5.9 by 1/3 $\sigma$, and 6.4 by SEM. The FACT-ES total score was 9.2 by 1/2 $\sigma$, 6.2 by 1/3 $\sigma$, and 6.7 by SEM, whereas those for FACT-ES TOI were 7.0, 4.6, and 5.3, respectively.

### Anchor-based MID estimates

The linear spline MID estimates by the anchor-based method are shown in **Table 3**. MID estimates of the FACT-B total scores were 15.9 for moderately worse, 7.9 for a little worse, 2.5 for a little better, and 4.9 for moderately better responses. Those of the FACT-ES total score were 16.7 for moderately worse responses, 8.3 for a little worse responses, 2.0 for a little better responses, and 3.9 for moderately better responses, whereas those for the FACT-ES TOI were 15.7 for moderately worse responses, 7.8 for a little worse responses, 1.5 for a little better responses, and 3.0 for moderately better responses. Furthermore, we categorized the patients

**Table 1. Patient characteristics at baseline (n = 47).**

| Variable | Value |
|---|---|
| Age (median and IQR) | 66 (61–72) |
| ECOG performance status, n (%) | |
| 0 | 39 (83.0) |
| 1 | 7 (14.9) |
| 2 | 1 (2.1) |
| Presence of visceral metastasis, n (%) | 23 (48.9) |
| Second-line endocrine therapy, n (%) | |
| Fulvestrant | 39 (83.0) |
| Selective estrogen receptor modulator | 4 (8.5) |
| Aromatase inhibitor | 1 (2.1) |
| mTOR inhibitor plus aromatase inhibitor | 3 (6.4) |
| Marital status, n (%) | |
| Married or with partner | 23 (48.9) |
| Never married | 5 (10.6) |
| Separated or divorced | 11 (23.4) |
| Widowed | 8 (17.0) |
| Education level, n (%) | |
| 9 years or less | 5 (10.6) |
| 10 to 12 years | 31 (66.0) |
| 13 years or more | 10 (21.3) |
| Missing | 1 (2.1) |
| Employment status at baseline, n (%) | |
| Full time | 4 (8.5) |
| Part time | 10 (21.3) |
| Homemaker | 16 (34.0) |
| Retired | 3 (6.4) |
| Unemployed | 14 (29.8) |
| Living situation (multiple answers allowed), n (%) | |
| Alone | 12 (25.5) |
| With husband | 24 (51.1) |
| With children | 18 (38.3) |
| With parents | 6 (12.8) |
| Others | 9 (19.1) |

IQR, interquartile range; ECOG, Eastern Cooperative Oncology Group; mTOR, mammalian target of rapamycin.

into three groups: decline (moderately and a little worse), stable (about the same), and improved (a little and moderately better). These three-category MID estimates based on the anchor-based method are demonstrated in **Table 4**. For FACT-B Total, FACT-ES Total, and FACT-ES TOI, the total scores were 7.3, 7.7, and 7.2 for decline; and 4.4, 4.1, and 3.4 for improvement, respectively. For all scores, MID estimates for decline were larger than the MID estimates for improvement by both the 3-category and linear spline methods.

## Relationship between clinical benefit and HRQOL scores or estimated MIDs of FACT-ES

The cumulative distribution functions for the FACT-ES total score according to the clinical benefit status at 1 and 3 months are shown in **Fig 2A and 2B**, respectively. Using the medians

**Table 2. Summary of HRQOL scores and MID estimates based on the distribution method.**

| | | | | MID estimate | | |
| --- | --- | --- | --- | --- | --- | --- |
| | Number of responded/ Number of expected | Mean (SD) | Cronbach's $\alpha$ | 1/2 SD | 1/3 SD | SEM |
| FACT-G total | | | | | | |
| Baseline | 47/47 (100%) | 75.1 (14.7) | 0.85 | 7.3 | 4.9 | 5.7 |
| 1 month | 43/47 (91.5%) | 74.9 (13.8) | 0.85 | 6.9 | 4.6 | 5.4 |
| 3 months | 43/47 (91.5%) | 74.6 (16.1) | 0.90 | 8.1 | 5.4 | 5.1 |
| Weighted mean | - | - | - | 7.4 | 5.0 | 5.4 |
| FACT-G TOI | | | | | | |
| Baseline | 47/47 (100%) | 41.5 (9.0) | 0.85 | 4.5 | 3.0 | 3.5 |
| 1 month | 44/47 (93.6%) | 41.9 (8.5) | 0.86 | 4.2 | 2.8 | 3.2 |
| 3 months | 43/47 (91.5%) | 40.8 (9.9) | 0.89 | 5.0 | 3.3 | 3.2 |
| Weighted mean | - | - | - | 4.6 | 3.0 | 3.3 |
| FACT-B total | | | | | | |
| Baseline | 47/47 (100%) | 100.2 (17.6) | 0.86 | 8.8 | 5.9 | 6.7 |
| 1 month | 43/47 (91.5%) | 100.4 (15.9) | 0.84 | 7.9 | 5.3 | 6.3 |
| 3 months | 43/47 (91.5%) | 99.8 (19.3) | 0.89 | 9.6 | 6.4 | 6.3 |
| Weighted mean | - | - | - | 8.8 | 5.9 | 6.4 |
| FACT-B TOI | | | | | | |
| Baseline | 47/47 (100%) | 66.6 (12.5) | 0.84 | 6.2 | 4.2 | 5.0 |
| 1 month | 44/47 (93.6%) | 67.1 (11.8) | 0.84 | 5.9 | 3.9 | 4.8 |
| 3 months | 43/47 (91.5%) | 66.1 (13.6) | 0.87 | 6.8 | 4.5 | 4.9 |
| Weighted mean | - | - | - | 6.3 | 4.2 | 4.9 |
| FACT-ES total | | | | | | |
| Baseline | 47/47 (100%) | 140.9 (18.2) | 0.85 | 9.1 | 6.1 | 7.1 |
| 1 month | 43/47 (91.5%) | 140.7 (17.0) | 0.85 | 8.5 | 5.7 | 6.6 |
| 3 months | 43/47 (91.5%) | 140.1 (20.4) | 0.90 | 10.2 | 6.8 | 6.4 |
| Weighted mean | - | - | - | 9.2 | 6.2 | 6.7 |
| FACT-ES TOI | | | | | | |
| Baseline | 47/47 (100%) | 107.3 (13.3) | 0.83 | 6.7 | 4.4 | 5.4 |
| 1 month | 44/47 (93.6%) | 107.3 (13.5) | 0.85 | 6.8 | 4.5 | 5.2 |
| 3 months | 43/47 (91.5%) | 106.3 (15.0) | 0.88 | 7.5 | 5.0 | 5.2 |
| Weighted mean | - | - | - | 7.0 | 4.6 | 5.3 |
| ESS | | | | | | |
| Baseline | 47/47 (100%) | 65.8 (7.9) | 0.77 | 4.0 | 2.6 | 3.8 |
| 1 month | 43/47 (91.5%) | 65.3 (8.0) | 0.78 | 4.0 | 2.7 | 3.8 |
| 3 months | 43/47 (91.5%) | 65.7 (7.9) | 0.79 | 4.0 | 2.6 | 3.7 |
| Weighted mean | - | - | - | 4.0 | 2.7 | 3.7 |

HRQOL, health-related quality of life; MID, minimally important difference; SD, standard deviation; SEM, standard error of measurement; FACT, Functional Assessment of Cancer Therapy; G, general; TOI, trial outcome index; B, breast; ES, endocrine symptom; ESS, endocrine symptom subscale.

The weighted mean was calculated using the number of scores at each time point as weight.

of the estimated distribution-based and anchor-based MIDs (8 for decline and 5 for improvement), the proportion of MID decline in the FACT-ES total score at 1 month was 20% for patients who experienced clinical benefit and 26.1% for the patients who did not (difference −6.1; 95% CI −31.2 to 19; P = 0.917). Those at 3 months were 9.1% and 23.8% (difference −14.7; 95% CI −36.5 to 7.1; P = 0.373). The proportions of MID improvement in the FACT-ES total score at 1 month were 40% for patients who experienced clinical benefit and 21.7% for

**Table 3. Linear spline MID estimates based on the anchor method.**

| | Estimate | Difference (vs stable) |
|---|---|---|
| FACT-G total | | |
| Moderately worse | -15.2 | -14.9 |
| A little worse | -7.7 | -7.5 |
| About the same | -0.3 | - |
| A little better | 2.9 | 3.1 |
| Moderately better | 6.0 | 6.3 |
| FACT-G TOI | | |
| Moderately worse | -13.6 | -14.0 |
| A little worse | -6.6 | -7.0 |
| About the same | 0.4 | - |
| A little better | 3.0 | 2.6 |
| Moderately better | 5.6 | 5.2 |
| FACT-B total | | |
| Moderately worse | -15.9 | -15.9 |
| A little worse | -8.0 | -7.9 |
| About the same | -0.1 | - |
| A little better | 2.4 | 2.5 |
| Moderately better | 4.9 | 4.9 |
| FACT-B TOI | | |
| Moderately worse | -14.4 | -14.9 |
| A little worse | -7.0 | -7.4 |
| About the same | 0.5 | - |
| A little better | 2.5 | 2.0 |
| Moderately better | 4.4 | 4.0 |
| FACT-ES total | | |
| Moderately worse | -16.6 | -16.7 |
| A little worse | -8.3 | -8.3 |
| About the same | 0.0 | - |
| A little better | 2.0 | 2.0 |
| Moderately better | 4.0 | 3.9 |
| FACT-ES TOI | | |
| Moderately worse | -15.1 | -15.7 |
| A little worse | -7.3 | -7.8 |
| About the same | 0.6 | - |
| A little better | 2.0 | 1.5 |
| Moderately better | 3.5 | 3.0 |
| ESS | | |
| Moderately worse | -8.2 | -8.2 |
| A little worse | -4.1 | -4.1 |
| About the same | 0.0 | - |
| A little better | -0.3 | -0.3 |
| Moderately better | -0.6 | -0.6 |

MID, minimally important difference; FACT, functional assessment of cancer therapy; G, general; TOI, trial outcome index; B, breast; ES, endocrine symptom; ESS, endocrine symptom subscale.

**Table 4. Three-category MID estimates based on the anchor method.**

| Three-category changes from baseline | The number of patients | | Estimate | Difference (vs stable) |
|---|---|---|---|---|
| | at 1 month | at 3 months | | |
| FACT-G total | | | | |
| Decline | 4 | 10 | -8.2 | -7.8 |
| Stable | 27 | 22 | -0.4 | - |
| Improve | 10 | 9 | 4.4 | 4.7 |
| FACT-G TOI | | | | |
| Decline | 4 | 10 | -7.1 | -7.3 |
| Stable | 28 | 22 | 0.3 | - |
| Improve | 10 | 9 | 4.2 | 4.0 |
| FACT-B total | | | | |
| Decline | 4 | 10 | -7.9 | -7.3 |
| Stable | 27 | 22 | -0.5 | - |
| Improve | 10 | 9 | 3.9 | 4.4 |
| FACT-B TOI | | | | |
| Decline | 4 | 10 | -6.8 | -6.8 |
| Stable | 28 | 22 | 0.0 | - |
| Improve | 10 | 9 | 3.8 | 3.8 |
| FACT-ES total | | | | |
| Decline | 4 | 10 | -8.2 | -7.7 |
| Stable | 27 | 22 | -0.5 | - |
| Improve | 10 | 9 | 3.6 | 4.1 |
| FACT-ES TOI | | | | |
| Decline | 4 | 10 | -7.2 | -7.2 |
| Stable | 28 | 22 | 0.0 | - |
| Improve | 10 | 9 | 3.4 | 3.4 |
| ESS | | | | |
| Decline | 2 | 3 | -4.1 | -4.1 |
| Stable | 38 | 32 | 0.0 | - |
| Improve | 2 | 5 | -0.5 | -0.5 |

MID, minimally important difference; FACT, functional assessment of cancer therapy, G: general; TOI, trial outcome index; B, breast; ES, endocrine symptom; ESS, endocrine symptom subscale.

n represents sum of the number of scores at 1 and 3 months.

patients who did not (difference 18.3%; 95% CI −9.0 to 45.6; P = 0.333). Those at 3 months were 31.8% and 28.6% (difference 3.2%; 95% CI −24.2 to 30.7; P = 1.0).

The mean changes in the FACT-ES total score from baseline according to the clinical benefit status are shown in **Fig 3**. The mean changes in the patients who experienced clinical benefit were 1.3 (95% CI −4.1 to 6.8) at 1 month and 2.2 (95% CI −1.2 to 5.6) at 3 months, whereas they were −4.2 (95% CI −7.9 to −0.5) at 1 month and −1.9 (95% CI −6.6 to 2.8) at 3 months in those who did not. The mean differences between the clinical benefit statuses were 5.5 (95% CI, 1.0–12.1; P = 0.099) at 1 month and 4.1 (95% CI -1.7–10.0; P = 0.162) at 3 months.

## Discussion

This study investigated the HRQOL of patients with MBC having a low sensitivity to ET. Among several QOL modules, we chose FACT-ES as a suitable PRO during ET. When interpreting HRQOL data, it is important to consider clinically meaningful absolute score

a
b

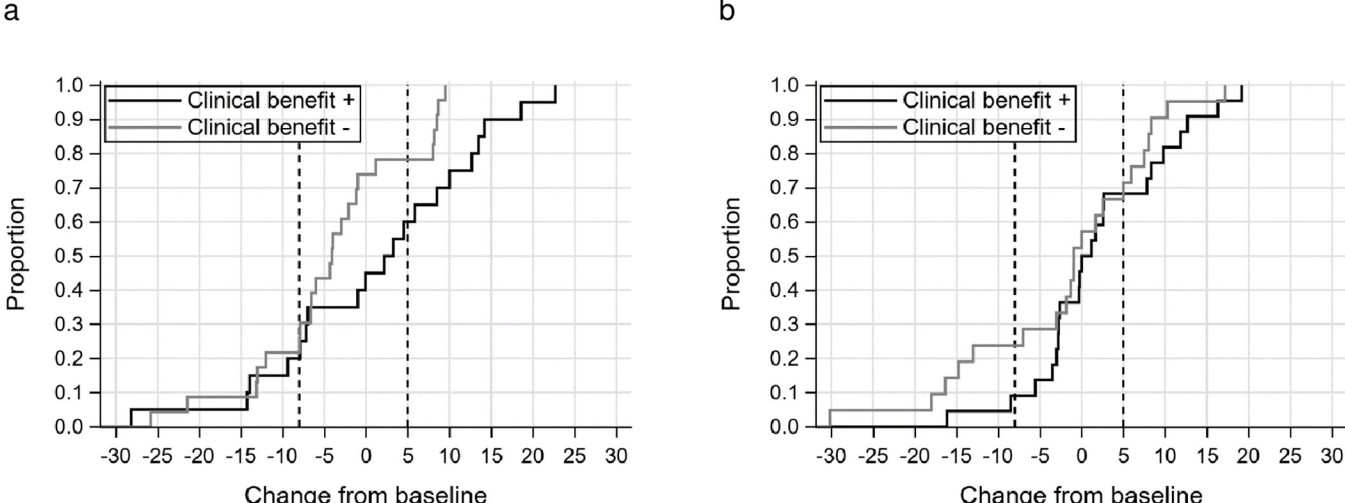

**Fig 2. FACT-ES score cumulative distribution functions by clinical benefit status at 1 month and 3 months.** (a) The cumulative distribution functions for the FACT-ES total score according to the clinical benefit status at 1 month. The vertical dotted lines indicate the MID thresholds. (b) The cumulative distribution functions for the FACT-ES total score according to the clinical benefit status at 3 months. The vertical dotted lines indicate the MID thresholds. FACT-ES, Functional Assessment of Cancer Therapy-Endocrine Symptoms; MID, minimally important difference.

differences. However, to date, neither the MIDs for FACT-ES nor those of the Japanese cohort for FACT-B have been investigated. Therefore, this is the first study in which the MIDs of the HRQOL score are estimated. MIDs are generally estimated using the distribution-based and

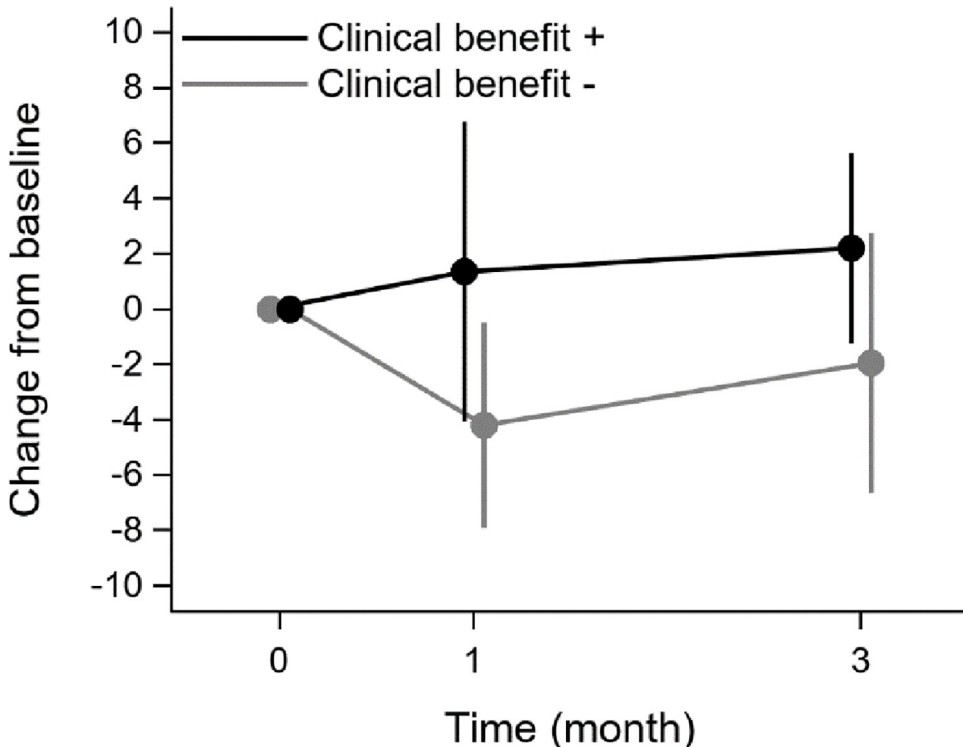

**Fig 3. The mean changes in the FACT-ES total score from baseline according to clinical benefit status.** FACT-ES, Functional Assessment of Cancer Therapy-Endocrine Symptoms.

anchor-based methods [12]. When using the anchor-based method, deterioration and improvement often have different values. In this study, as described in another report [19], the magnitude of the decline of all the HRQOL scores was larger than that of the improvement of the HRQOL scores. One reason for this discrepancy is 'response bias [18]'. Specifically, many cancer patients tend to feel improvement even in small changes, as they do not want to think that the time and cost of treatment were wasted.

Regarding HRQOL evaluation for patients with MBC during second-line ET, some studies have shown an improvement in the HRQOL for patients treated with a combination of ET and targeted therapy [20–22]. However, there is no HRQOL study for patients with MBC having a low sensitivity to ET during second-line ET focusing on endocrine symptoms. Moreover, there is no data on Japanese patients. Therefore, to discuss the differences in hormone sensitivity and cross-cultural discrepancy in FACT scores, we compared our data with that of previous literature. First, the baseline HRQOL scores in this study were similar to those of the FALCON study, a randomized controlled trial of first-line ET for MBC [23]. Conversely, the baseline score was better than that of the CONFIRM trial, which is a trial of second-line ET for MBC [24]. The reason for this is that more than half of the patients had no visceral metastasis or previous ET did not result in prolonged adverse events. Second, regarding the cross-cultural differences of MID, distribution- and anchor-based MIDs in our data were also similar to those in previous studies [14, 25]. Based on the abovementioned points, we speculate that MIDs of FACT scores are generally within the same range across different treatment settings, cultures, and ethnicities. However, these were comparisons between different trials and should be interpreted with caution. Further large-scale global studies are needed to validate this specific population.

For further analyses of the relationship between clinical response and HRQOL, we defined MIDs of the FACT-ES total score as 8 and 5 points for declined and improved HRQOL scores based on the distribution- and anchor-based method, respectively. The proportions of patients who experienced MID change in the FACT-ES total score did not differ with the clinical benefit status. However, the proportion of patients with improved MIDs tended to be higher at 1 month in those with clinical benefit than in those without. Further, the proportion of patients with MID deterioration tended to be lower at 3 months in those who experienced clinical benefit than in those who did not. Although the results showed no statistically significant association between MIDs and CBR, the patients who experienced clinical benefit tended to have fewer declines and much more improvement. Moreover, the mean change in the FACT-ES total score from baseline at 1 and 3 months also tended to improve in patients who experienced a clinical benefit. In general, even if clinical benefit is achieved with cytotoxic chemotherapy, the adverse events may deteriorate the HRQOL. In contrast, since there are fewer side effects of ET, it is conceivable that clinical efficacy correlates with HRQOL (e.g., non-hematological adverse events of more than grade 3 in this study were very rare, such as fatigue in two patients, depression in one patient, or appetite loss in one patient). We also reported that changes in HRQOL were associated with treatment alterations in patients with MBC in a substudy of a large randomized trial, which might have affected progression-free survival (PFS) and overall survival [26]. Given the 7-month PFS in this study, we hypothesized that HRQOL decline within 3 months would be a surrogate for disease progression. Future research will reveal the appropriate timing for HRQOL assessments to predict treatment efficacy. This evidence integrating HRQOL and efficacy provides important information for decision-making in the treatment of patients with MBC.

This study had some limitations. First, the sample size was small, and we could not estimate the MIDs in detail based on the seven SSQ categories. Second, the planned duration of HRQOL surveys was relatively short (only 3 months) to avoid missing data due to long-term

survey. Changes in HRQOL that appeared late after three months and the impact of long-lasting side effects of ET were not evaluated in this study. Third, the MIDs of FACT-ES were estimated for a limited cohort, such as patients on second-line ET or those mostly treated with fulvestrant. Additionally, the current standard of care is the combination of ET with cyclin-dependent kinase 4/6 (CDK 4/6) inhibitors. Fourth, the relationship between OS and HRQOL was unclear because the primary outcome of the main study was CBR. Finally, statistical analyses were conducted based on several assumptions. In anchor-based MID analyses, we assumed that the trend was linear for both decline and improvement. This linear trend assumption was considered reasonable because there were only three data points (i.e., a little, moderately, and very much) for decline and improvement in the SSQ. We also assumed that the HRQOL scores were completely missing at random. Although it was unclear whether this assumption was fulfilled in this study, we had only 3–4 missing scores at each point, which was unlikely to have a substantial impact on the results. However, this study stands out in that the MIDs of FACT-ES were estimated using the SSQ prospectively, which has not been published before. Further, we have reaffirmed the importance of assessing HRQOL during the treatment of patients with MBC. Given that the addition of CDK4/6 inhibitors did not degrade HRQOL in the previous study [22], we believe this data is also applicable in the current treatment setting.

## Conclusions

In this study, there were no apparent cultural differences in the FACT scores between Japanese patients and patients from other countries. For the first time in literature, the MIDs of the FACT-ES total score were estimated to be 5 for improvement and 8 for decline. Patients who were able to maintain their HRQOL during the first 3 months from the start of second-line ET were likely to experience clinical benefits. Therefore, evaluating FACT-ES may serve as a surrogate for the effectiveness of second-line ET for patients with MBC having a low sensitivity to ET. Based on our results, early imaging examination and transition to the next sequential therapy such as chemotherapy may be indicated in patients with decreased FACT-ES scores. Further studies with larger sample sizes and current standards of care, such as ET with some targeted therapies are warranted.

## Supporting information

**S1 Checklist. TREND statement checklist.**
(DOC)

**S1 File. The names of the ethics committees/institutional review boards.**
(DOCX)

**S1 Protocol. Study protocol (English version).**
(PDF)

**S2 Protocol. Study protocol (Japanese version).**
(PDF)

## Acknowledgments

This study was conducted as a research support project of the General Incorporated Association of Comprehensive Support Project for Oncological Research of Breast Cancer (CSPOR-BC).

We thank Drs. T. Yamaguchi and T. Shimoi for their valuable comments on this paper.

## Author Contributions

**Conceptualization:** Yuichiro Kikawa, Tomomi Fujisawa, Reiki Nishimura, Naruto Taira.

**Data curation:** Yuichiro Kikawa, Yasuhiro Hagiwara.

**Investigation:** Kazuhiro Araki, Takayuki Iwamoto, Takafumi Sangai, Tadahiko Shien, Shintaro Takao, Reiki Nishimura, Masato Takahashi, Tatsuya Toyama, Tomohiko Aihara, Hirofumi Mukai, Naruto Taira.

**Project administration:** Tatsuya Toyama, Tomohiko Aihara, Hirofumi Mukai.

**Supervision:** Naruto Taira.

**Writing – original draft:** Yuichiro Kikawa.

**Writing – review & editing:** Yuichiro Kikawa, Yasuhiro Hagiwara, Tomomi Fujisawa, Kazuhiro Araki, Takayuki Iwamoto, Takafumi Sangai, Tadahiko Shien, Shintaro Takao, Reiki Nishimura, Masato Takahashi, Tatsuya Toyama, Tomohiko Aihara, Hirofumi Mukai, Naruto Taira.

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
