## [Decision Letter · Decision Letter 0]

7 Sep 2022

PONE-D-22-12031Health-related quality of life and estimation of the minimally important difference in the Functional Assessment of Cancer Therapy-Endocrine Symptom score in postmenopausal ER+/HER2- metastatic breast cancer with low sensitivity to endocrine therapyPLOS ONE

Dear Dr. Kikawa,

Thank you for submitting your manuscript to PLOS ONE. After careful consideration, we feel that it has merit but does not fully meet PLOS ONE’s publication criteria as it currently stands. Therefore, we invite you to submit a revised version of the manuscript that addresses the points raised during the review process.

The manuscript has been evaluated by two reviewer, and their comments are available below.

The reviewers raised a number of major concerns that need attention. They request additional information on regarding the statistical methods used in this study. The reviewers also request more information regarding background and rationale.

Could you please revise the manuscript to carefully address the concerns raised?

We look forward to receiving your revised manuscript.

Kind regards,

Jamie Royle

Staff Editor

PLOS ONE

Journal Requirements:

2. Thank you for including your ethics statement:  "All procedures were performed in accordance with the Helsinki declaration and the institutional review board at each study site approved the final protocol. Written informed consent was obtained from all study participants".   

a. For studies reporting research involving human participants, PLOS ONE requires authors to confirm that this specific study was reviewed and approved by an institutional review board (ethics committee) before the study began. Please provide the specific name of the ethics committee/IRB that approved your study, or explain why you did not seek approval in this case.

YK received honoraria from Eisai, Novartis, Pfizer, Lilly, Taiho, and Chugai, outside the submitted work. MT received grants from Kyowa Hakko Kirin and Taiho, personal fees from AstraZeneca, Eisai, Eli Lilly, and Pfizer, outside the submitted work. TT received grants and personal fees from Chugai, Eisai, Novartis, Takeda, Nippon Kayaku, Pfizer, Lilly, and Daiichi Sankyo, and personal fees from AstraZeneca, and grants from Taiho and Kyowa Kirin, outside the submitted work. HM received honoraria from Pfizer, Takeda, Daiichi Sankyo, and Taiho, and grants from the Japanese government, Daiichi Sankyo, and Pfizer, outside the submitted work. The remaining authors have no conflicts of interest to disclose.

5. We note that the original protocol that you have uploaded as a Supporting Information file contains an institutional logo. As this logo is likely copyrighted, we ask that you please remove it from this file and upload an updated version upon resubmission.

Reviewers' comments:

Reviewer's Responses to Questions

**Comments to the Author**

1. Is the manuscript technically sound, and do the data support the conclusions?

Reviewer #1: Yes

Reviewer #2: Partly

2. Has the statistical analysis been performed appropriately and rigorously? 

Reviewer #1: Yes

Reviewer #2: No

3. Have the authors made all data underlying the findings in their manuscript fully available?

Reviewer #1: Yes

Reviewer #2: Yes

4. Is the manuscript presented in an intelligible fashion and written in standard English?

Reviewer #1: Yes

Reviewer #2: Yes

5. Review Comments to the Author

Reviewer #1: This is a very interesting study of the impacts of endocrine therapy on quality of life for patients with endocrine-resistant advanced breast cancer. The investigators should be congratulated for this effort. However, there are some issues that need to be addressed before the manuscript can be considered for publication in our journal.

MAJOR COMMENTS:

• In the Background (page 5), I would strongly encourage the authors to mention also CDK4/6 inhibitors as a current standard of care for patients whose breast cancer is progressing/relapsing on endocrine therapy. This can further improve treatment outcomes compared with endocrine therapy alone.

• In the Methods (page 7), I wonder whether the authors could expand on the rationale for monitoring QoL at 1 and 3 months after baseline? This is a very short period of time, and it may not be appropriate in the context of the timing of disease responses (typically seen after 3 months of therapy) and side effects trends seen on endocrine therapy (that may settle after the first 3-4 months of therapy). This is clearly a key limitation of the study, as it would have been very important to document QoL outcomes and impacts of endocrine therapy also later on (eg, 6 and 12 months) also in patients switching to further systemic therapy options.

MINOR COMMENTS:

• In the Background (page 5), I wonder whether the authors could define “low sensitivity to initial ET” within the HORSE-BC study. Was this defined based on ER Allred score or based on outcomes on endocrine therapy (eg, PFI)? I believe this is clarified in the Methods, but it would be useful to outline this also in the Background for additional clarity from the very beginning of the manuscript.

• In the Background (page 5-6), please consider expanding on the data available on QoL in this specific setting. Is there any evidence documenting QoL outcomes in patients with advanced breast cancer receiving endocrine therapy, CDK4/6 inhibitors or chemotherapy that would lend further support to this analysis? These studies should be referenced.

• In the Background (page 6), the authors should also mention the challenges in delivering the relevant questionnaires (eg, rates of completion and missing data) as a limitation of QoL analyses in this setting.

• In the Methods (page 7), please consider removing the summary of findings of the HORSE-BC study as readers can read these in the main study paper and in order to save words here.

• In the Discussion (page 14), please re-phrase and clarify the following sentence “as they value the duration and cost of treatment”.

• In the Discussion (page 14-15), I would caution about comparing the findings of different trials (HORSE-BC, FALCON and CONFIRM). This should be clearly stated in the Discussion.

• In the Discussion (page 15), the hypothesis of considering HR-QoL as a surrogate of PFS is a key aspect for the discussion. I would strongly encourage the authors to expand on this. Is there any evidence on this topic? How can we validate HR-QoL as a surrogate of survival outcomes. What are the directions for future research on this aspect? This would make a compelling case for increasing integration of QoL in decision-making and as a co-primary endpoint of clinical trials investigating novel treatments in the palliative setting.

Reviewer #2: The study aims to clarify the clinically significant scores of the HR-QOL as MIDs associated with ET for MBC and to determine the effect of secondary ET on the HRQOL in postmenopausal patients with MBC that was less sensitive to primary ET using the proportion of patients without a decline in the MID.

The manuscript can be further improved based on the following comments.

HR-QOL to be written as HRQOL throughout the manuscript.

Page 9, the sentence ‘The association between the responses to the SSQs and the HR-QOL score changes’ requires revision.

Data Analysis

The statistical software including publisher name and version and the acceptance level of statistical significance to be stated.

For Fisher’s exact test, one or two-tailed to be stated.

A statement on the assumptions of GEE was fulfilled is to be provided. The sample size was less than 50 and whether the correction method was applied is to be stated.

Results

Table 2, the word MID estimate to be placed in central between 1/2Sd=D, 1/3 SD, and SEM

Page 11, ‘and 6.4, respectively, by SEM’ to be written as ‘and 6.4 by SEM respectively’.

Page 12, ‘FACT-B, FACT-ES, and FACT-ES TOI’ to be written as FACT-B Total, FACT-ES Total, and FACT-ES TOI.

Page 12, for the statement ‘The 3-category MID estimates for decline and improvement were close to slightly worse and moderately better MID estimates by the linear spline method, respectively’ how the word ‘close’ is defined?

Page 13, for the statement ‘The differences between clinical benefit statuses were 5.5 (95% CI 1.0 to 12.1) at 1 month and 4.1 (95% CI −1.7 to 10.0) at 3 months (P = 0.066 for pooled group difference)’ the word mean difference, p-value at 1 month and p-value at 3 months to be provided. It is not clear what ‘P = 0.066 for pooled group difference’ refers to.

Discussion

Page 14, for ‘as described in another report [15], The magnitude’ T for The to be in small cap.

6. PLOS authors have the option to publish the peer review history of their article (what does this mean?). If published, this will include your full peer review and any attached files.

Reviewer #1: **Yes: **Nicolò Matteo Luca Battisti

Reviewer #2: No

---

## [Author Response · Author response to Decision Letter 0]

27 Sep 2022

Dear Managing Editor:

We thank the reviewers for carefully reading our manuscript and providing useful feedback.

Our responses to the reviewers’ comments are provided below.

Reviewer #1:

In the Background (page 5), I would strongly encourage the authors to mention also CDK4/6 inhibitors as a current standard of care for patients whose breast cancer is progressing/relapsing on endocrine therapy. This can further improve treatment outcomes compared with endocrine therapy alone.

Response to Reviewer #1

Thank you for your suggestion. We have added the following text to the revised manuscript in the background section.

On page 5, lines 7-9

Currently, the combination of ET plus cyclin-dependent kinase (CDK) 4/6 inhibitors is the standard second-line therapy for patients who have developed resistance to initial ET alone, based on the results of trials showing prolonged survival.

To further clarify that this study did not include patients who received CDK4/6 inhibitors, the following underlined statement has been added.

On page 5, lines 13-15

Therefore, we conducted a multicenter prospective observational cohort study (HORSE-BC study) including patients with HR-positive and HER2-negative MBC who were considered as having a low sensitivity to initial ET before CDK 4/6 inhibitors were available in Japan.

In the Methods (page 7), I wonder whether the authors could expand on the rationale for monitoring QoL at 1 and 3 months after baseline? This is a very short period of time, and it may not be appropriate in the context of the timing of disease responses (typically seen after 3 months of therapy) and side effects trends seen on endocrine therapy (that may settle after the first 3-4 months of therapy). This is clearly a key limitation of the study, as it would have been very important to document QoL outcomes and impacts of endocrine therapy also later on (eg, 6 and 12 months) also in patients switching to further systemic therapy options.

Response to Reviewer #1

Thank you for your invaluable comment.

We decided to evaluate HRQOL within 3 months in consideration of the increase in missing data because we hypothesized that the clinical benefit rate for 6 months would be up to 50%. However, as the reviewer pointed out, changes in efficacy and HRQOL beyond 3 months are important and the lack of those data was a key limitation of our study. Therefore, we have added the following text as a limitation of the study.

On page 22, lines 3-4

Changes in HRQOL that appeared late after three months and the impact of long-lasting side effects of ET were not evaluated in this study.

In the Background (page 5), I wonder whether the authors could define “low sensitivity to initial ET” within the HORSE-BC study. Was this defined based on ER Allred score or based on outcomes on endocrine therapy (eg, PFI)? I believe this is clarified in the Methods, but it would be useful to outline this also in the Background for additional clarity from the very beginning of the manuscript.

Response to Reviewer #1

Thank you for your suggestion.

We followed the definitions for primary and secondary resistance to endocrine therapy proposed in the ESO-ESMO guidelines to define sensitivity to the subsequent endocrine therapy (see protocol page 30).

As suggested by the reviewer, we have added the following underlined text to the revised manuscript.

On page 5, lines 11-13

especially for patients assumed to have a low sensitivity to ET, such as those who relapsed during the first 2 years of adjuvant ET or had disease progression within 6 months of first-line ET as described in the ESO-ESMO Guidelines.

In the Background (page 5-6), please consider expanding on the data available on QoL in this specific setting. Is there any evidence documenting QoL outcomes in patients with advanced breast cancer receiving endocrine therapy, CDK4/6 inhibitors or chemotherapy that would lend further support to this analysis? These studies should be referenced.

In the Background (page 6), the authors should also mention the challenges in delivering the relevant questionnaires (eg, rates of completion and missing data) as a limitation of QoL analyses in this setting.

Response to Reviewer #1

Thank you for your invaluable comment.

In the review article (Lancet Oncol. 2018 Sep;19(9):e459-e469), Madeline Pe et al. of the EORTC identified a lack of standardization of analysis despite many clinical trials evaluating the quality of life in advanced metastatic breast cancer. Moreover, as the reviewer commented, missing data would be an unavoidable problem in HRQOL assessment. The review by Pe et al. found that many studies did not describe the handling of missing data, which is problematic. We have added the following underlined text to the revised manuscript and cited the article as a reference.

On page 6, lines 6-8

However, the methods of analysis and handling of missing data have not been standardized [10]. Further, whether statistically significant differences are clinically important may be unclear.

In the Methods (page 7), please consider removing the summary of findings of the HORSE-BC study as readers can read these in the main study paper and in order to save words here.

Response to Reviewer #1

Thank you for your comment. We have removed the following text from the manuscript.

On page 7

In summary: 1) overall, 56 patients were enrolled between February 2016 and January 2017, of which 49 were analyzed; 2) the median age was 66 years; 3) 41 patients (82%) received fulvestrant, 5 patients (10%) received selective estrogen receptor modulators, 3 patients (6%) received a mammalian target of rapamycin inhibitor plus steroidal aromatase inhibitor (AI), and 1 patient (2%) received AI alone; 4) the overall CBR was 44.9% (90% confidence interval [CI] 34.6 to 57.6, P = 0.009); and 5) the median progression-free survival (PFS) was 7.1 (95% CI 5.6 to 10.6) months.

However, we have mentioned the period of enrollment to clarify that the study was conducted prior to the time when CDK4/6 inhibitors were available.

On page 7, lines 13-14

Overall, 56 patients were enrolled between February 2016 and January 2017, of whom 49 were analyzed. The study results have been described previously [7]

In the Discussion (page 14), please re-phrase and clarify the following sentence “as they value the duration and cost of treatment”.

Response to Reviewer #1

Thank you for your comment.

We have changed this to the following sentence to clarify the meaning.

On page 19, lines 19 to page 20, line 1

as they do not want to think that the time and cost of treatment were wasted.

In the Discussion (page 14-15), I would caution about comparing the findings of different trials (HORSE-BC, FALCON and CONFIRM). This should be clearly stated in the Discussion.

Response to Reviewer #1

Thank you for your invaluable comment.

We have added the following text to the revised manuscript.

On page 20, lines 14-16

However, these were comparisons between different trials and should be interpreted with caution. Further large-scale global studies are needed to validate this specific population.

In the Discussion (page 15), the hypothesis of considering HR-QoL as a surrogate of PFS is a key aspect for the discussion. I would strongly encourage the authors to expand on this. Is there any evidence on this topic? How can we validate HR-QoL as a surrogate of survival outcomes. What are the directions for future research on this aspect? This would make a compelling case for increasing integration of QoL in decision-making and as a co-primary endpoint of clinical trials investigating novel treatments in the palliative setting.

Response to Reviewer #1

Thank you for pointing this out.

Recently, we published an important paper that highlighted an association between QOL and treatment alteration (Support Care Cancer. 2022 Jul 20). Regarding its surrogacy, however, the time gap significantly varied between the survival event and the last assessment of QOL. Further research is needed to establish the optimal assessment point to predict the survival events. Frequent assessment using ePRO might be beneficial. Based on the above points, we have added the following statement and have cited our study to the revised manuscript.

On page 21, lines 12-18

We also reported that changes in HRQOL were associated with treatment alterations in patients with MBC in a sub-study of a large randomized trial, which might have affected progression-free survival (PFS) and overall survival [26]. Given the 7-month PFS in this study, we hypothesized that HRQOL decline within 3 months would be a surrogate for disease progression. Future research will reveal the appropriate timing for HRQOL assessments to predict treatment efficacy. This evidence integrating HRQOL and efficacy provides important information for decision-making in the treatment of patients with MBC.

Reviewer #2:

HR-QOL to be written as HRQOL throughout the manuscript.

Response to Reviewer #2

Thank you for your suggestion. We have changed the term to HRQOL.

Page 9, the sentence ‘The association between the responses to the SSQs and the HR-QOL score changes’ requires revision.

Response to Reviewer #2

Thank you for your suggestion.

We have rephrased this to the following sentence.

On page 9, line 17

Responses to the SSQ and the corresponding HRQOL score change values 

Data Analysis

The statistical software including publisher name and version and the acceptance level of statistical significance to be stated.

For Fisher’s exact test, one or two-tailed to be stated.

Response to Reviewer #2

Thank you for your suggestion. 

We have added the following text to the revised manuscript.

On page 10, lines 15-17

All statistical analyses were performed using the SAS software (version 9.4; SAS Institute Inc., Cary, NC, USA). All p-values were two-sided. Differences were considered statistically significant at p < 0.05, without a multiplicity adjustment.

A statement on the assumptions of GEE was fulfilled is to be provided. The sample size was less than 50 and whether the correction method was applied is to be stated.

Response to Reviewer #2

We thank the reviewer for these comments.

Several assumptions were made in the GEE analyses. The main assumptions included: the correct mean structure in the marginal model and missing completely at random assumption. Unfortunately, we cannot, in general, verify that these assumptions were fulfilled from observed data (although we can verify that these assumptions were NOT fulfilled from data). Therefore, we have included these assumptions as a limitation in the revised manuscript. 

The sandwich variance estimator tends to underestimate standard error in small-sample longitudinal studies. However, a simulation study showed that when continuous outcomes from 40 to 50 patients were analyzed, substantial underestimation of standard error did not occur (https://pubmed.ncbi.nlm.nih.gov/26585756/). In addition, balanced data from small number of visits do not tend to lead to underestimated standard error ("Applied longitudinal analysis 2nd ed" by Fitzmaurice, Laird, and Ware). We only included two visits in this study. therefore, we did not make small-sample correction to the sandwich variance estimator in generalized estimating equation analyses.

Based on the above, we have added the following text to the revised manuscript.

On page 10, line 17 to page 11, line 1

We did not make small-sample corrections to the sandwich variance estimator in generalized estimating equation analyses because more than 40 patients were included and HRQOL scores from only two visits were analyzed.

On page 22, line 17 to page 23, line 5

Finally, statistical analyses were conducted based on several assumptions. In anchor-based MID analyses, we assumed that the trend was linear for both decline and improvement. This linear trend assumption was considered reasonable because there were only three data points (i.e., a little, moderately, and very much) for decline and improvement in the SSQ. We also assumed that the HRQOL scores were completely missing at random. Although it was unclear whether this assumption was fulfilled in this study, we had only 3–4 missing scores at each point, which was unlikely to have a substantial impact on the results.

Table 2, the word MID estimate to be placed in central between 1/2Sd=D, 1/3 SD, and SEM

Response to Reviewer #2

Thank you for your suggestion. We have corrected Table 2.

Page 11, ‘and 6.4, respectively, by SEM’ to be written as ‘and 6.4 by SEM respectively’.

Response to Reviewer #2

Thank you for your suggestion. We have corrected the text on page 13, line 7.

Page 12, ‘FACT-B, FACT-ES, and FACT-ES TOI’ to be written as FACT-B Total, FACT-ES Total, and FACT-ES TOI.

Response to Reviewer #2

Thank you for your suggestion. We have corrected the text on page 15, line 6

Page 12, for the statement ‘The 3-category MID estimates for decline and improvement were close to slightly worse and moderately better MID estimates by the linear spline method, respectively’ how the word ‘close’ is defined?

Response to Reviewer #2

Thank you for your invaluable comment. We have removed the sentence because it was not appropriate for the results.

Page 13, for the statement ‘The differences between clinical benefit statuses were 5.5 (95% CI 1.0 to 12.1) at 1 month and 4.1 (95% CI −1.7 to 10.0) at 3 months (P = 0.066 for pooled group difference)’ the word mean difference, p-value at 1 month and p-value at 3 months to be provided. It is not clear what ‘P = 0.066 for pooled group difference’ refers to.

Response to Reviewer #2

Thank you for pointing this out.

We have removed the pooled group difference to clarify that the results at two time points were presented. Subsequently, we have added the mean and two P values.

On page 19, lines 2-4

The mean differences between the clinical benefit statuses were 5.5 (95% CI, 1.0–12.1; P = 0.099) at 1 month and 4.1 (95% CI -1.7–10.0; P = 0.162) at 3 months.

Discussion

Page 14, for ‘as described in another report [15], The magnitude’ T for The to be in small cap.

Response to Reviewer #2

Thank you for noticing our mistake. We have corrected it.

---

## [Decision Letter · Decision Letter 1]

15 Nov 2022

Health-related quality of life and estimation of the minimally important difference in the Functional Assessment of Cancer Therapy-Endocrine Symptom score in postmenopausal ER+/HER2- metastatic breast cancer with low sensitivity to endocrine therapy

PONE-D-22-12031R1

Dear Dr. Kikawa,

We’re pleased to inform you that your manuscript has been judged scientifically suitable for publication and will be formally accepted for publication once it meets all outstanding technical requirements.

Kind regards,

Justin C. Brown

Section Editor

PLOS ONE

Additional Editor Comments (optional):

Reviewers' comments:

Reviewer's Responses to Questions

**Comments to the Author**

1. If the authors have adequately addressed your comments raised in a previous round of review and you feel that this manuscript is now acceptable for publication, you may indicate that here to bypass the “Comments to the Author” section, enter your conflict of interest statement in the “Confidential to Editor” section, and submit your "Accept" recommendation.

Reviewer #1: All comments have been addressed

Reviewer #2: All comments have been addressed

2. Is the manuscript technically sound, and do the data support the conclusions?

Reviewer #1: Yes

Reviewer #2: Partly

3. Has the statistical analysis been performed appropriately and rigorously? 

Reviewer #1: Yes

Reviewer #2: Yes

4. Have the authors made all data underlying the findings in their manuscript fully available?

Reviewer #1: Yes

Reviewer #2: Yes

5. Is the manuscript presented in an intelligible fashion and written in standard English?

Reviewer #1: Yes

Reviewer #2: Yes

6. Review Comments to the Author

Reviewer #1: Thank you for addressing my suggestions. The manuscript is now suitable for publication in our journal.

Reviewer #2: (No Response)

7. PLOS authors have the option to publish the peer review history of their article (what does this mean?). If published, this will include your full peer review and any attached files.

Reviewer #1: **Yes: **Nicolò Matteo Luca Battisti

Reviewer #2: No

---

## [Editor Report · Acceptance letter]

18 Nov 2022

PONE-D-22-12031R1 

Health-related quality of life and estimation of the minimally important difference in the Functional Assessment of Cancer Therapy-Endocrine Symptom score in postmenopausal ER+/HER2- metastatic breast cancer with low sensitivity to endocrine therapy 

Dear Dr. Kikawa:

I'm pleased to inform you that your manuscript has been deemed suitable for publication in PLOS ONE. Congratulations! Your manuscript is now with our production department. 

Kind regards, 

on behalf of

Dr. Justin C. Brown 

Section Editor

PLOS ONE